# Interventions for combating COVID-19 misinformation: A systematic realist review

Robert Dickinson[1]*, Dominique Makowski[2], Harm van Marwijk[1], Elizabeth Ford[1]

**1** Department of Primary Care and Public Health, Brighton and Sussex Medical School, Brighton, United Kingdom, **2** Department of Psychology, University of Sussex, Brighton, United Kingdom

* r.dickinson@sussex.ac.uk

## Abstract

Misinformation is a growing concern worldwide, particularly in public health following the COVID-19 pandemic in which misinformation has been attributed to tens of thousands of unnecessary deaths. Therefore a search for effective interventions against misinformation is underway, with widely varying proposed interventions, measures of efficacy, and groups targeted for intervention. This realist systematic review of proposed interventions against COVID-19 misinformation assesses the studies themselves, the characteristics and effectiveness of the interventions proposed, the durability of effect, and the circumstances and contexts within which these interventions function. We searched several databases for studies testing interventions published from 2020 onwards. The search results were sorted by eligibility, with eligible studies then being coded by themes and assessed for quality. Thirty-five studies were included, representing eight types of intervention. The results are promising to the advantages of game-type interventions, with other types scoring poorly on either scalability or impact. Backfire effects and effects on subgroups were reported on intermittently in the included studies, showing the advantages of certain interventions for subgroups or contexts. No one intervention appears sufficient by itself, therefore this study recommends the creation of packages of interventions by policymakers, who can tailor the package for contexts and targeted groups. There was high heterogeneity in outcome measures and methods, making comparisons between studies difficult; this should be a focus in future studies. Additionally, the theoretical and intervention literatures need connecting for greater understanding of the mechanisms at work in the interventions. Lastly, there is a need for work more explicitly addressing political polarisation and its role in the belief and spread of misinformation. This study contributes toward the expansion of realist review approaches, understandings of COVID-19 misinformation interventions, and broader debates around the nature of politicisation in contemporary misinformation.

**Data availability statement:** All relevant data are within the paper and its Supporting Information files.

**Funding:** The author(s) received no specific funding for this work.

**Competing interests:** The authors have declared that no competing interests exist.

## Introduction

Misinformation has been a societal issue throughout history [1]. This phenomenon can be seen in many areas, but perhaps most clearly in public health, where alongside the introduction of many major advances in medicine came movements of resistance and misinformation [2]. In the contemporary, systemic misinformation is a well-established by-product of increasing reliance on the internet and social media for the dissemination of news and information [3,4]. Public concern about misinformation appeared to reach a new height in 2016 in relation to the US Presidential Election, particularly around perceptions of misinformation campaigns supporting Donald Trump's bid for the presidency [5,6]. By many accounts, this period resulted in the development of an infrastructure of misinformation, accelerated by social media algorithms to reach new and greater audiences [7,8]. In 2020, when the COVID-19 pandemic began, misinformation began and continued to punctuate the public understanding of the pandemic and the public health response thereto [9].

For the purposes of this study, misinformation is broadly defined as misleading news, media content, or other information which directs viewers towards understandings of the world that do not align with a socio-politically derived consensus of legitimacy, which may be intentional or unintentional, consciously or unconsciously biased, and may or may not contain mechanisms of psychological manipulation. This definition arises out of a growing literary shift towards viewing misinformation as "that which contradicts the best expert evidence available at the time" [10].

The pre-COVID misinformation intervention landscape appears to focus on fact-checking [11–15]. This is not to suggest that this literature exclusively focused on fact-checking, as a rich literary body nonetheless existed examining with depth and alacrity understandings of the effectiveness and appeal of misinformation based in cognitive ability, emotional appeal, partisanship, sensationalism, and fear-mongering [14–20]. Fact-checking can be described as a form of debunking in which information is retroactively checked for veracity and if found to be inaccurate changed. Importantly, an additional step in fact-checking and other forms of debunking is attempting to reach the audience initially exposed to the misinformation and retroactively change their internalised understanding of the information [21]. Recently, accuracy nudges have been championed as a new, primary intervention-type [14]. Accuracy nudges refer to a variety of interventions that 'nudge' people to consider the veracity of the information they are seeing or are about to see. This can include prompts that appear on-screen next to links to news articles, or fact-checks that appear alongside social media posts but can also take on a wide variety of forms. Below is an example from X/Twitter highlighted by a red circle (Fig 1), that shows crowdsourced fact-checking to appear next to suspected misinformation [22].

Although championed through seminal studies like [23], accuracy nudge interventions have since garnered significant criticism on their effectiveness and the potential impact of partisan bias in participants [24–26] including replication studies that did not replicate the initial findings [27].

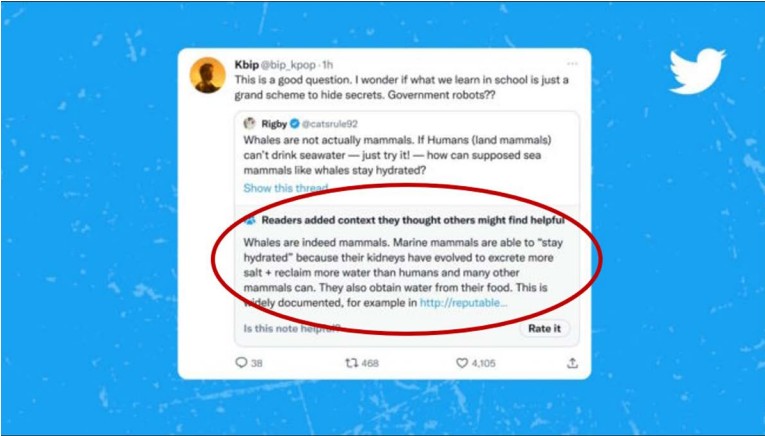

**Fig 1. Example of an accuracy nudge.**

In the theoretical literature, discussion of misinformation interventions focused on inoculation, backfire effects, and the importance of worldview in intervention effectiveness [14,15,17,19]. Inoculation refers to the idea of priming people before they might encounter misinformation to make them more aware of it with the goal of building resilience against it. The 'backfire' effect is an issue widely theorised about in the literature around misinformation, typically centred on the idea that an intervention seeking to combat misinformation might end up reinforcing 'in-group' thinking among those most conspiracy-minded or most politically polarised [12,14,28]. For these people, it is speculated that an intervention (e.g., labeling their favoured sources as false or untrustworthy) could further entrench them in their distrust of legitimate public health messaging. This has the potential to make the intervention not only less effective, but potentially negative in impact. This is known as the 'backfire effect' and will be evaluated in the included studies. A concept arising from the policy and psychology disciplines that could contribute to addressing potential backfire effects is framing [29,30]. Framing refers to the use of strategic messaging that is created with the intent of aligning with the extant worldview of the target audience to make new ideas or information as congruent as possible. In practice, framing has been found to improve fact-checking and accuracy nudge interventions [31].

There are studies testing interventions, and many reviews of the theory surrounding misinformation, but as yet no reviews attempting to achieve a broader overview and evaluation of the various interventions which emerged in the COVID-19 context. This project aims to contribute both toward the expansion and application of realist review approaches, while simultaneously contributing toward better understandings of interventions against COVID-19 misinformation. As COVID-19 continues to spread and the possibility of a new pandemic lurks as an ever-present threat, developing the best understanding of interventions to effectively combat COVID-19 misinformation will serve to help prepare policymakers and public health apparatuses for the next pandemic.

**Research Question: Which interventions are most effective in combating spread of and belief in COVID misinformation?**

**Sub-questions:**

**RQ1: Which types of interventions work best?**

**RQ2: Which groups of people do they work for?**

**RQ3: Under which circumstances are the interventions most effective?**

**RQ4: What is the quality of studies testing interventions to combat spread of and belief in misinformation?**

## Theoretical framework

This study takes the theoretical framing of a realist review, focusing on uncovering the mechanisms that explain how interventions produce effects in specific contexts, working to ensure that findings can be generalized or applied to similar circumstances. These mechanisms function as the underlying processes that drive change in an intervention. They could be cognitive, as is the case for accuracy nudge interventions, where a brief reminder provides a cognitive 'nudge' to temporarily boost cognition when clicking on a link or news article. They could be social, as is the case for community engagement interventions, where extended and deep interaction with a community works to identify and slowly change misinformed group narratives. They could also be behavioural, as is the case for game interventions, where the act of playing a game itself is the intervention, and the interaction between player and game functions to educate and inoculate against misinformation. Table 1a in the results section provides a description of the mechanism at work within each included study for this review.

The other key piece of a realist review puzzle, contexts, are the environmental, cultural, economic, or other situational factors that influence how, why, where, and when interventions are effective. For some interventions like accuracy nudges, their use is limited to digital spaces, fitting contexts like social media platforms. For others like educational or message framing interventions, fitting contexts for use might include difficult to reach or distrustful communities. For debunking interventions, context becomes a question of time and popularity, best fitting situations where misinformation has already emerged and spread. Table 2 in the results section, alongside the wider 'Context and Generalisability' subsection of the results explains in greater detail the contextual nature of the interventions included in this review, as well as ranking them by generalisability.

By together clarifying the mechanisms and contexts for the interventions included, this review provides a deeper understanding of how the interventions operate and offers a theoretical contribution to realist review methods in public health and psychology. By utilising a realist review approach in the study of misinformation, where little such examination currently exists, this study contributes to the expansion of realist review approaches. Similarly, as both realist reviews and misinformation are quickly growing and relatively newly expanding areas of literature, this approach offers new insights into the study of misinformation interventions, particularly in the case of COVID-19. This review also highlights the importance of context in the countering of misinformation. By demonstrating the limited generalisability of many interventions, it contributes to the ongoing refinement of misinformation interventions to best allow bespoke policy packages of interventions to be developed best fitting the needs of the communities involved.

**Table 1. Interventions sorted by type.**

| Intervention Type | Example | Mechanism | Key Findings |
|---|---|---|---|
| Accuracy Nudges | 'Community notes' on Twitter | Prompts to encourage reflection on information accuracy | Mixed effectiveness, minimal impact on those heavily misinformed |
| Community Engagement | Series of workshops with community leaders | Co-created interventions tailored to community needs | Most impactful, but low scalability |
| Debunking | Fact-checking | Correcting misinformation after exposure | Moderate effectiveness, strongest among older populations |
| Prebunking | Pre-emptive messages warning about misinformation | Inoculation against misinformation before exposure | Effective, but mainly for those with healthy information habits |
| Education | A comic teaching about misinformation | Informational campaigns (videos, infographics, comics) | Highly variable impact based on preexisting knowledge levels |
| Games | Playing by taking the role of the misinformer online | Interactive misinformation training (e.g., "Bad News" game) | High impact and scalability, best for digital-literate audiences |
| Message Framing | Reshaping communication to fit different worldviews | Strategic messaging tailored to audience worldview | Effective but risk of backfire effects in certain subgroups |

**Table 2.** Study characteristics table.

| Study | Intervention-type | Quality (/8) | Mechanism | Delivery Method | methodology | n= | Outcome |
|---|---|---|---|---|---|---|---|
| Abroms et al. 2024 | education | 7 | moderated social media discussions | moderator posts | longitudinal RCT | 478 | improvement in vaccination intention |
| Agley et al. 2021 | education | 7 | infographic | viewing as part of the trial | two-arm, parallel-group, RCT | 1017 | small effect but highly scalable |
| Amazeen et al. 2022 | prebunking | 6 | inoculation messages | self-administered online survey | inoculation messages | 540 | only among those with healthy attitudes |
| Amin et al. 2021 | accuracy nudges | 3 | stimulating attention | Visual Selective Attention System tool | experiment | 38 | decision behaviour improved |
| Arechar et al. 2023 | accuracy nudges | 7 | four digital literacy tips | trial assignment | RCT | 34286 | broadly effective' |
| Armand et al. 2024 | message framing | 7 | framing, interaction | phone | RCT | 2916 | increased vaccine acceptance and trust |
| Aslett et al. 2022 | accuracy nudges | 7 | source credibility labels | embedded in feeds and search results | randomized field experiment | 3337 | no significant change |
| Basol et al. 2020 | games | 6 | the game | trial | replication and extension experiment | 196 | significantly improves veracity judgment |
| Bender et al. 2023 | message framing | 4 | framing | physician presenting information via video | randmized 2x2 between-subject design | 652 | Small but significant impact |
| Cook et al. 2024 | games | 8 | the game | trial assignment | RCT | 829 | significant vaccination attitude improvement |
| DeGarmo et al. 2022 | community engagement | 6 | community outreach | community health promoters | randomized control trial | 1841 | significant, medium size effect |
| Dias et al. 2020 | accuracy nudges | 5 | logo banner | trial, but presented in facebook format | survey experiment | 6987 | no, even potentially counterproductive |
| Freeman et al. 2021 | message framing | 6 | framing | trial provision of written information | single-blind, parallel-group, RCT | 15014 | effective on the most vaccine-hesitant |
| Fung et al. 2022 | education | 4 | education | telephone | multi-week educational intervention | 25 | significant educational improvements |
| Gavin et al. 2022 | accuracy nudges | 4 | nudge | online trial survey | replicating studies in other regions | 2581 | lowered willingness to share misinformation |
| Iles et al. 2022 | message framing | 8 | framing | online trial assignment | randomized online experiment | 1804 | significant reduction in vaccine-hesitancy |
| Jiang et al. 2022 | prebunking | 7 | inoculation messages | trial assignment reading | 3 phase between-subject experiment | 123 | generated resistance to misinformation |
| Johnson et al. 2022 | education | 6 | real social media | trial assignment to watch videos | RCT | 842 | significant success compared to a control |
| Kreps et al. 2022 | accuracy nudges | 5 | false tags | trial assignment modeled after Facebook | survey experiment | 2000 | little effect on veracity judgment or sharing |
| Li et al. 2024 | message framing | 8 | framing | trial assignment | RCT | 226 | improved participant understanding |
| Ma et al. 2023 | games | 4 | the game | trial assignment | multi-study RCT | 311 | better discernment of misinformation |
| Maertens et al. 2021 | games | 8 | the game | trial assignment | longitudinal experiments | 515 | lasting increase in misinformation discernment |
| Maragh-Bass et al. 2022 | community engagement | 3 | digital storytelling | workshop developing them, then sharing | community workshops, storytelling | 11 | Suggests efficacy for marginalised communities |
| Msunyaro et al. 2023 | community engagement | 2 | community outreach | community health promoters | community health campaign | 87380 | contributed to improve COVID-19 vaccine coverage |

*(Continued)*

**Table 2.** (Continued)

| Study | Intervention-type | Quality (/8) | Mechanism | Delivery Method | methodology | n= | Outcome |
|---|---|---|---|---|---|---|---|
| Offer-Westort et al. 2024 | multiple | 8 | variety | facebook messenger | RCT | 15292 | heterogeneity in best intervention |
| Pennycook et al. 2020 | accuracy nudges | 7 | accuracy reminder | reminder at beginning of study | RCT | 1709 | accuracy nudges are simple and effective |
| Piltch-Loeb et al. 2022 | prebunking | 5 | inoculation messages | video | quasi-experimental, with control | 1991 | significant effects compared to control |
| Rasmussen et al. 2022 | education | 8 | educational videos | video | experiment | 6785 | decreased misinformation sharing |
| Spalvins et al. 2024 | message framing | 8 | framing | trial assignment | RCT | 187 | no significance found |
| Ugarte & Young 2023 | community engagement | 6 | peer leaders | group chats within private facebook groups | two-arm, parallel-group, RCT | 120 | lowers spread of misinformation |
| Van Stekelenburg et al. 2021 | education | 7 | infographic | trial assignment | longitudinal survey | 1202 | did not significantly improve belief accuracy |
| Vandormael et al. 2021 | education | 7 | educational video | social media distribution internationally | RCT | 15163 | boosted preventative knowledge |
| Veletsianos et al. 2022 | education | 1 | educational comic | trial assignment to read the comic | post-test only non-experimental design | 295 | comic was effective and engaging |
| Vijaykumar et al. 2021 | debunking | 5 | corrective information | trial assignment | two mixed-design experiments | 1454 | enhanced trust and sharing of accurate information |
| Yousuf et al. 2021 | debunking | 5 | debunking video | trial assignment to watch the video | randomized trial | 980 | significantly stronger rejection of misconceptions |

## Methodology

This review follows the Preferred Reporting Items for Systematic Reviews and Meta-Analyses [37] checklist, available in the supporting information. The review followed a pre-registered protocol submitted before the study began with PROSPERO, registration number CRD42023440580, record title: "Realist review: assessing intervention effectiveness in combating COVID misinformation", available at the PROSPERO website. Amendments to the information provided in the protocol were centred on the elimination of an initially-planned research question on the intersection of theory and intervention literatures within the reviewed studies. This research question was removed after data extraction and analyses revealed a dearth of theoretical investigation in the reviewed studies. Instead this lack of theoretical involvement in the reviewed studies is noted in the discussion.

## Search strategy

This review included a systematic search of Web of Science, Scopus, ASSIA, Psycinfo, and Pubmed to identify English language articles written between January 1, 2020 and June 22, 2023 performed following a pre-registered protocol conforming to the Preferred Reporting Items for Systematic Reviews and Meta-Analysis (PRISMA) statement [32]. An updating secondary search was performed using the same search strategy and methodology bringing this study up-to-date through 09 October 2024. Search strategy followed the protocol using pre-determined search terms, with results imported into Excel sheets for ease of deselection. Duplicates were removed and then an initial title-based screening was performed. Screening then followed based on abstract and then full-text review. Additional searching among the references of the included studies followed. Duplicate screening was performed by a team member (K.G.) on ~15% of studies

through all screening stages, with any disagreement resolved via discussion. Inter-rater agreement was found to be very high (~92%).

The full search-string chosen for this review, which was only applied to Titles and Abstracts, is as follows: (conspirac* OR anti-vax* OR anti-vaccine OR 'anti vaccine' OR misinform* OR fake OR fals*) AND (messag* OR rumor* OR argu* OR rhetoric OR spread*) AND (COVID OR COVID-19 OR coronavirus OR 'corona virus' OR pandemic*) AND interven*

## Eligibility

Trials or experimental studies were eligible if they were focused on reducing the spread of and vulnerability to COVID-19 misinformation in their participants, and tested an intervention meant to combat COVID-19 misinformation. Studies were required to be in the English language and been published between 2020 and 2024 as searching before the COVID-19 pandemic began was unnecessary.

## Quality assessment

The methodological quality of each study chosen for inclusion was assessed via Kennedy et al.'s [33] risk of bias tool for assessing study rigor. It includes eight items for appraisal: (1) cohort, (2) control or comparison group, (3) pre-post intervention data, (4) random assignment of participants to the intervention, (5) random selection of participants for assessment, (6) follow-up rate of 80% or more, (7) comparison groups equivalent on sociodemographics, and (8) comparison groups equivalent at baseline on outcome measures. This assessment tool was used for its flexibility regarding type of methods and interventions in the studies being assessed. Although this analysis was performed, no studies were excluded due to quality as realist reviews explicitly disagree with exclusion from quality concerns as explained below.

## Data extraction and analyses

The following information from included studies was extracted into a table to highlight study characteristics as can be seen in the next section: Study, intervention-type, 'working ingredient', 'delivery method', country of origin, methodology, number of participants, and whether the intervention was found to be successful. Additionally, a variety of other information was extracted to inform the other tables and charts found in the results section. All text from the eligible studies was imported into NVivo Pro 14 and the methods, results, and discussion sections underwent qualitative coding. Coding was done iteratively to categorise the findings. This iterative process evolved into a developed framework as the coding took place. For instance, if one intervention was identified from an article during coding, the coder attempted to assign it to a category within the emerging intervention framework. New subcategories were created if the current categories were insufficient, until all interventions were categorised. As coding progressed, the intervention framework came to be populated through the included studies. The heterogeneity of the included studies and their respective measures disallowed quantitative meta-analysis.

Regarding effectiveness, impact per participant and scalability were the primary variables analysed. Impact per participant refers to the level of individual behavioural change experienced by the participants of each intervention reviewed, as all were centred on individual behaviour. Scalability is a more complex variable consisting of several combined factors including generalisability (how effectively can results be replicated in other contexts and with other groups), resource-intensity (how expensive is the intervention in terms of time, money, and overall resource expenditure), and capacity for upscaling (how many people it could reach). With impact and scalability thus defined, effectiveness can be then analysed by how many people could be impacted and to what extent per person. A sub-analysis of context was undertaken by comparing context by intervention type, and laying out which participant groups were targeted by the interventions. Additional analysis was performed to investigate context beyond the community of participants within the intervention.

It is important to define 'circumstances' as used in RQ3. Here, circumstances refers to the context within which an intervention is taking place - such as geographic location, identities and wealth of the targeted community, and

structural and institutional factors within which the community and intervention will take place. Additionally, circumstances refers to the experience, resources, and capacity of the research team or implementing body performing the intervention.

## Results

### Study characteristics

35 papers were found that met inclusion criteria, including 6 out of reviewing the bibliographies of the 29 studies found through the search strategy described above. 636 were initially resulting from the searches, with 230 duplicate results removed, 341 deselected by title, and 45 deselected by full-text review, resulting in 26 eligible papers (Fig 1). A new search of the same parameters and strategy took place on 09 October 2024 to update the study, whose papers are incorporated into this results section. These new papers build on the existing themes and reinforce the conclusions in this study. All new papers fit into the categories previously established in the original search, and their conclusions on impact, scalability, and contextual groups and circumstances for which the interventions would excel are all in line and supportive of the studies from the original search. The proposed interventions in the most recent studies continue to be heterogeneous and highlight the need for flexible reviews that can analyse this variance.

Eligible papers were published between 2020 and 2024, with a variety of national, regional, and international participant groups and study origination countries. The papers reviewed utilised participant groups coming mainly from the USA through private research participant companies like MTurk, Lucid, Prolific, Pollfish, and YouGov but also targeted audiences within the US like essential workers [34] and 'Latinx' communities [35]. Beyond the US, participant groups from Germany, the UK, Hong Kong, China, Canada, the Netherlands, Brazil, Kyrgyzstan, India, and internationally were included in the reviewed studies. These studies split into the intervention framework developed in the data extraction process as follows: 8 studies using Accuracy Nudges; 8 using education; 3 using Prebunking; 4 using Games; 6 using message framing; 4 using Community Engagement; and 2 using Debunking (Table 1, Table 2).

Full details of the studies can be seen in the supporting information.

All eligible studies underwent quality assessment using Kennedy et al.'s [33] risk of bias tool for assessing study rigor, results are shown in S1 Table. Many studies lost several points due to lack of follow-up elements or not giving information on whether comparison groups were equivalent on demographics or baseline outcome measures. Iles et al. [36] and Maertens et al. [37] were the only perfect scoring studies in the initial search, with Veletsianos et al. [38] on the other side scoring only 1/8 as the lowest score of the assessed studies. In the updating search, the new studies in general scored higher than the initial batch – a promising sign for the literary body. When sorted into intervention-types, the average quality scores are relatively similar for each group, indicating a similar level of quality across the intervention-types. Additionally, the division of studies by intervention type allows us to mitigate the impact of any one study's limitations by synthesising across multiple studies within each intervention category. Further details on quality per included study are available in S1 Table.

### Intervention characteristics

The studies in this review tested interventions with far greater heterogeneity than the dominant interventions proposed before the COVID-19 pandemic (accuracy nudges and fact-checking). As can be seen above in the study characteristics table, the studies were iteratively sorted into intervention-types as laid out in the methodology section. These intervention-types included: accuracy nudges, community engagement, debunking, prebunking, education, games, and message framing. This section will briefly introduce these intervention types and their defining characteristics.

Accuracy nudges in the reviewed studies consisted of mechanisms including: stimulating attention [39], source credibility labels [40], logo banners to help identify trustworthiness of sources [41], accuracy reminders [23], and tags that mark information as false [42]. These various intervention mechanisms fit under accuracy nudges due to their common

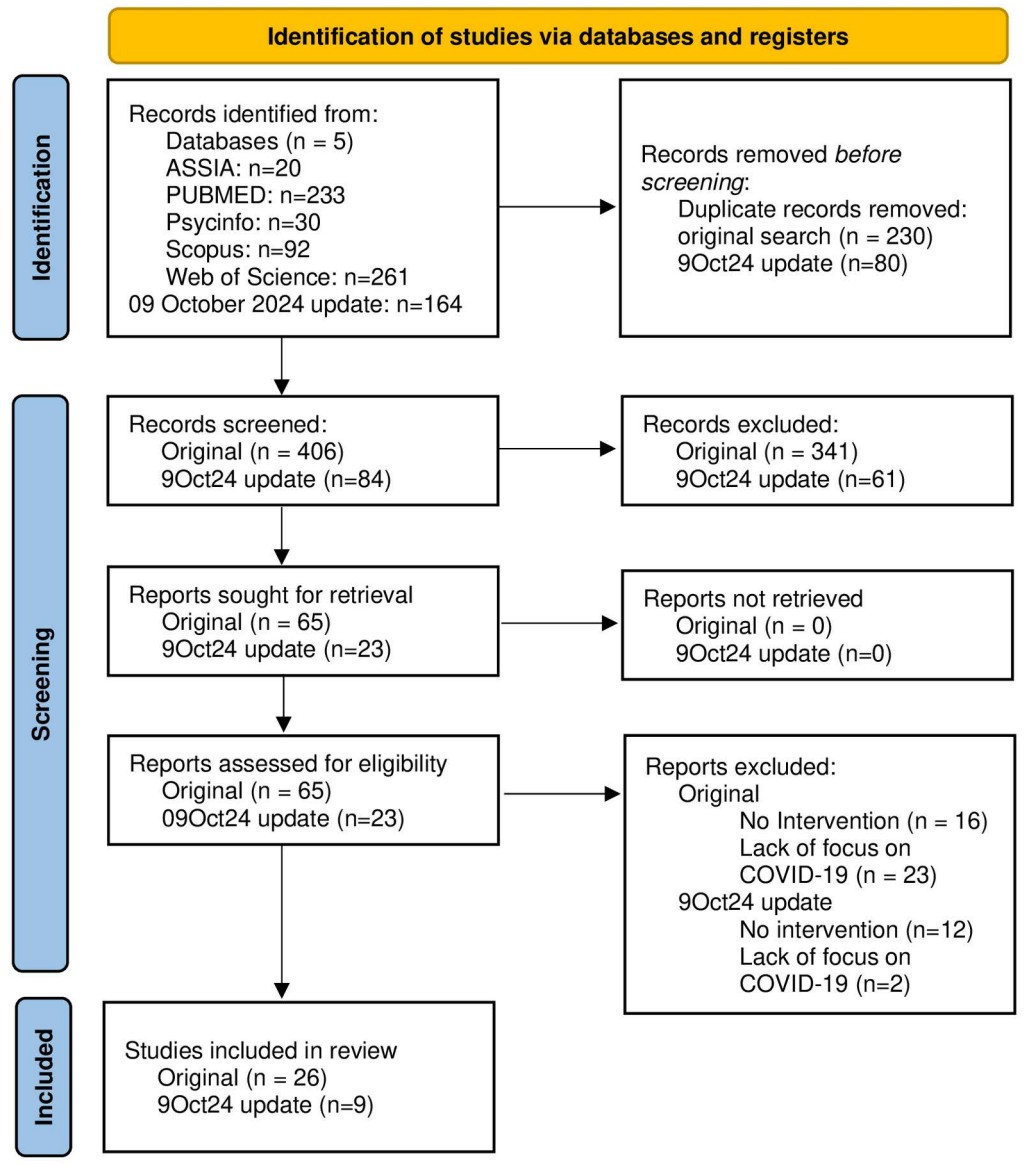

**Fig 2. PRISMA flow diagram for systematic reviews.**

characteristics as simple, fast, attention-grabbing labels or reminders that 'nudge' the participant to consider information veracity and bring that consideration into the forefront of their minds immediately before reading the information.

Community engagement is difficult to characterise by intervention mechanism because the defining aspect of community engagement occurs *before* intervention mechanism is determined in the research design. Instead of pre-determining intervention mechanisms and delivery methods, community engagement involves co-creation of the intervention alongside and in collaboration with the targeted community, to be bespoke to the unique context and circumstances of the community [34,35,43].

Debunking refers broadly to reactive interventions (e.g., fact-checking) that seek to 'debunk' existing misinformation and help people exposed to it rethink their belief and formulate new understandings of the relevant information [44,45]. In

contrast, 'prebunking' interventions seek to build resilience to misinformation in people preemptively before exposure has occurred, and potentially even before the piece of misinformation has been created/spread [46]. This typically takes the form of inoculation messages administered to participants before exposure to potential misinformation. In this way they are similar to accuracy nudges – the key difference being that prebunking is more extensive than accuracy nudges. The inoculation messages are more significant, take longer to process, and are intended to take the full attention of the participant for the duration of the message, whereas accuracy nudges are fast and often involve the periphery of a participant's attention. In the reviewed studies characterised as prebunking, all three involve inoculation messages as their intervention mechanism [28,47,48].

Education is the most heterogeneous of the intervention-types and can be difficult to categorise as educating the participant is essential to all interventions working to address misinformation. In the reviewed studies, this intervention-type involved mechanisms such as: videos [49], comics [38], infographics [50,51], and a multimodal intervention using authentic social media messaging [52]. The defining characteristic of the reviewed studies in this intervention-type is the primacy and exclusivity of education as the goal of the intervention. For instance, in Vandormael et al. [49], an educational video was released and distributed internationally with the goal of maximising viewership, but with no additional features of the intervention beyond watching the video.

Game intervention-types are characterised by the inclusion of a computer game for participants to play as the primary intervention-mechanism. This can be seen in all three of the included studies under this categorisation. These games inform players (participants) on the tactics and manipulation used to create and spread misinformation, with the goal of creating an inoculation effect and helping bolster veracity-judgment in participants. For example, Bad News is the name of the game used in Maertens et al. [37], a popular game used in many studies outside the purview of this review as well. In this game, players take on the role of an antagonist, creating misinformation and working to spread it through social media and the internet.

Message framing as an intervention-type is characterised by the use of psychological framing in the development of the language used in the intervention. Whether presenting information via video or written information, what distinguishes these studies as message framing is their strategic use of language to attempt to make their information transfer to participants as congruent with their extant worldview as possible. This then helps participants internalise that information effectively and can address intervention design concerns around potential backfire effects.

## Intervention effectiveness

The two variables most central to answering which interventions work 'best' appear to be scalability and impact. If impact is too low, the intervention might not actually engender sufficient behavioural change in participants to combat the misinformation. Similarly, if an intervention cannot be upscaled, it has no capacity to address COVID-19 misinformation at a systemic level. The 'ladder' visuals below represent the intervention-types relative to one another across these two variables (Figs 3–4). Relative impact is determined by the measured impact on participants in each study. These measures are not consistent, yet with the authors' interpretations, comparisons are possible. These relative measurements focus on the impact per participant, with no regard to number of participants or scalability. Inversely, the scalability ladder visual focuses on scalability with no regard to impact per participant. These visuals are meant to simplify and ease understanding of the results, and are purely relative.

On the bottom rung of impact per participant is accuracy nudges, whose impact is heavily debated. Some authors in this review such as Pennycook et al. [23], champion this intervention type and claim significant impact in their results. Gavin et al. [27], who replicated Pennycook et al. [23], found mixed results that stood at odds with the original study. Amin et al. [39] found impact on decision behaviour and tendency to share misinformation with their study, but the rest of the studies in this intervention group found either minimal impact [42], only impact on certain groups [40], or no impact [41] who even noted potential counterproductivity.

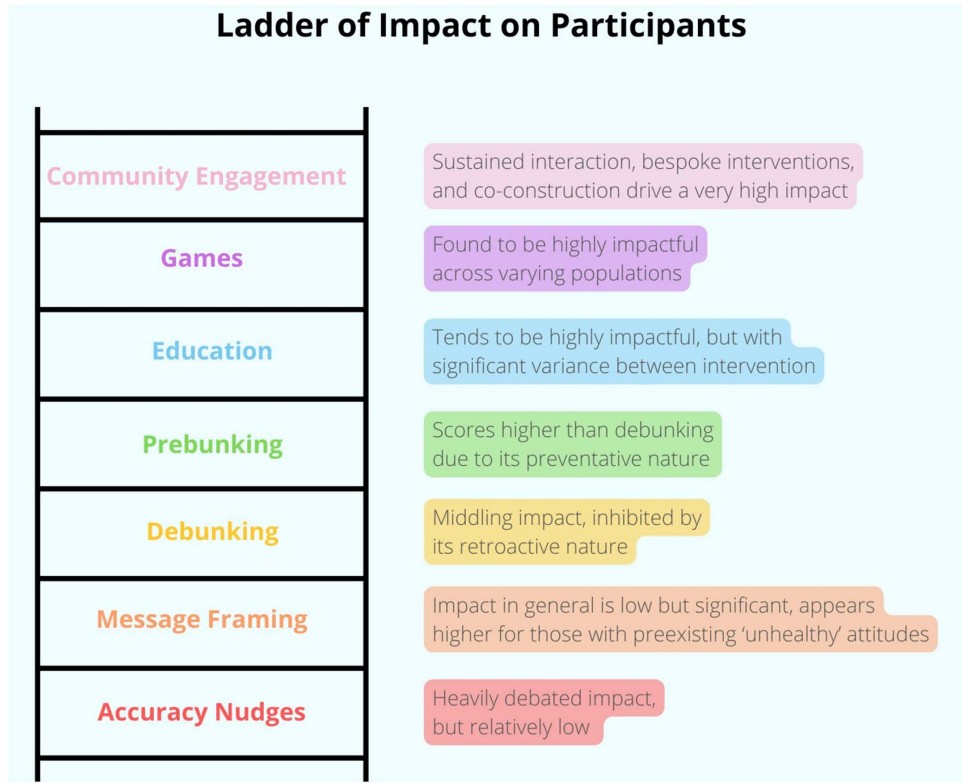

**Fig 3. Ladder of Impact on Participants.**

Framing of public health messages is next along the impact ladder. Studies testing interventions using different framings of public health messaging found significant impact [36,53,54], although not as high as other intervention-types included in this review. This impact was largely reserved for those heaviest consumers of misinformation and those most vaccine-hesitant [53].

Debunking by its nature must occur retroactively, which limits impact as the initial exposure must be overcome. In this way, debunking has two independent goals: to both disprove internalised misinformation and convince the participant of the veracity of legitimate information. This is a barrier to impact, which is noted by both Vijaykumar et al. [44] and Yousuf et al. [45]. Vijaykumar et al. [44] found no impact on perception or willingness to share misinformation yet found enhanced credibility and readiness to share accurate information because of their intervention. However, Yousuf et al. [45] found that exposure to their intervention did result in enhanced trust in government and significantly stronger rejection of vaccination misconceptions.

Prebunking, as the preventative version of debunking, scores better on impact. Prevention is found to be more powerful in a variety of aspects than reactive debunking. All three included studies [28,47,48] found significant impact among participants, although in the case of Amazeen et al. [47] this significance was limited to those with preexisting 'healthy' attitudes. Impact on participants was found to include generating resilience against misinformation, less willingness to share misinformation, and greater willingness to receive a vaccine.

Education is the most varied type of intervention with a range of impact between the individual tested interventions (the relative score here is an aggregate). At best, educational interventions have the potential to be a form of systematic prebunking with great effectiveness. In the reviewed studies, they were found to improve knowledge and increase resilience

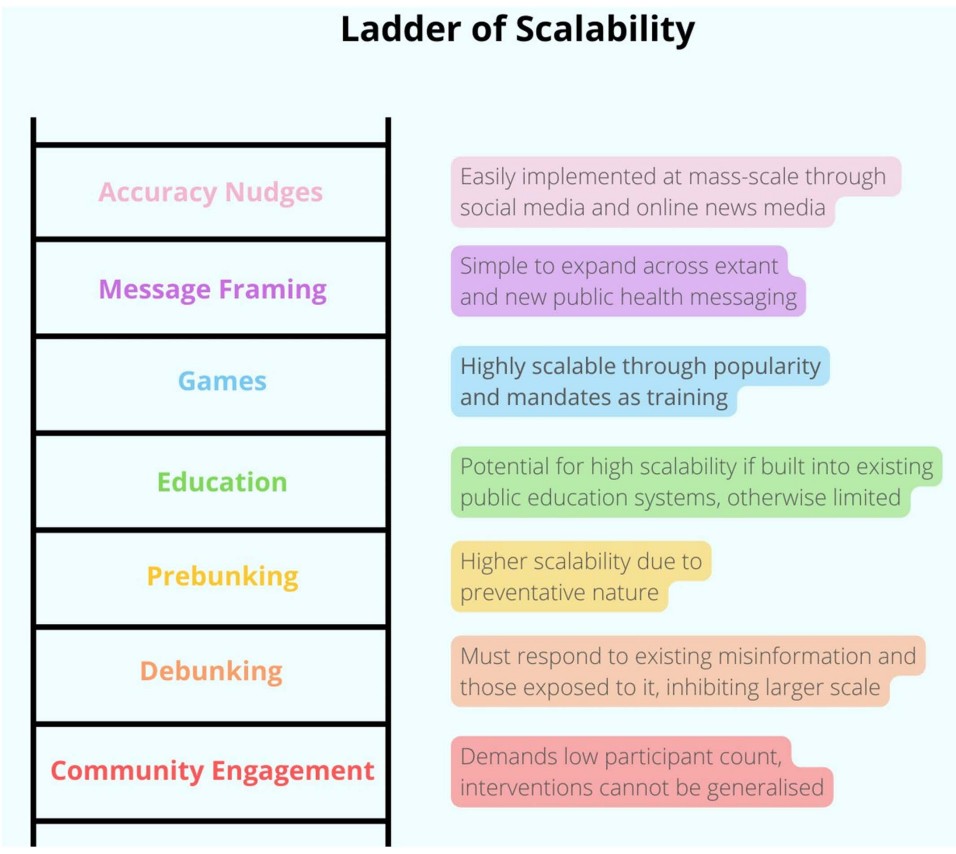

**Fig 4. Ladder of scalability.**

to misinformation at significant levels, particularly among populations with low preexisting knowledge levels [49,52]. However, Van Stekelenburg et al. [51] found no significant impact, highlighting the variability of this intervention-type.

Games were consistently found to be highly impactful across the various populations who played them, with high levels of durability and longevity compared to other intervention-types reviewed and significant impact levels for all kinds of preexisting attitudes towards vaccination and COVID-19 [37,46,55]. Every reviewed study found significant impact, which corresponds with a high relative impact score, although still below the bespoke and prolonged interventions within community engagement.

Community engagement is the single most impactful intervention-type, with sustained interaction and bespoke interventions to specifically targeted communities who then themselves are brought into the intervention process and invited to participate, make their voices heard, and have their concerns addressed in a bespoke, personal, and trusted manner. All reviewed studies found significant and extensive impact among their participants.

Community engagement is essentially impossible to scale upwards. It inherently requires small numbers of participants and high levels of resource and time investment by those implementing the intervention. The interventions themselves are then not even intended to be generalisable, but rather bespoke to and befitting the contextual needs of the community involved. Community engagement can only effectively be done at a small scale over long periods of time involving the building of trust with community, the proactive engagement and co-creation of interventions and implementation strategies with the community itself, and the implementation strategies themselves can take years to accomplish [34,43].

Debunking scores quite low in scalability. Debunking at a large scale is extremely difficult as it is inherently based on preexisting misinformation and cannot effectively prevent additional misinformation. Further, it must always attempt to reach those specific populations initially exposed to the targeted misinformation to be 'debunked', which is difficult and resource-intensive.

Prebunking does not need to attempt to find and target those who already saw some misinformation as the intervention occurs before misinformation is seen. For this reason, prebunking is easier to scale upwards than debunking and is relatively lower in resource-cost. Implementation of prebunking involves the development of 'inoculation messages' [28,47] as written messages or video content intended to raise resiliency of participants against misinformation.

Without being built into the public education system, educational interventions may struggle to scale upwards, relying on peer educational champions [56] or social media 'virality' to spread [49]. Adjusting anything within the public education system is highly resource-intensive, even though those changes are then highly impactful and wide-reaching. However, when performed in smaller scale as in the included interventions, educational interventions can be substantially reduced in resource intensity [49].

Although not as easily scalable as message framing or accuracy nudges, games are nonetheless relatively highly scalable when compared to the other intervention-types in this review. As the games are already developed, introducing them to new populations is then relatively simple, resource-inexpensive, and quick.

Message framing has high scalability with the simple addition of language strategising and purposeful narrative framings applied to extant and new public health messaging. Message framing is only slightly more resource intensive than accuracy nudges in that it must be bespoke to particular narratives, communities, and groups. However, in each bespoke circumstance, still the resource intensity would be low.

Accuracy nudges is undeniably the highest scalability intervention-type. The core reason why accuracy nudges are so scalable is the extremely low resource intensity needed to implement them. It requires the insertion of nudges in social media feeds and news articles. This would be easy and inexpensive for social media corporations and newspapers to implement, even when scaled into the extreme levels of interaction and users involved in contemporary social media.

## Durability of effect

In the studies that did test for longevity/durability of impact in their tested intervention, consistent low levels were found, with findings indicating high reliance on intervention repetition and regular testing of misinformation resilience over a sustained (and potentially indefinite) period to reach functional durability of effect. The study that looked most closely at this was Maertens et al. [37] which performed one of the only longitudinal studies included in this review explicitly investigating longevity of impact using the 'Bad News' game as its chosen intervention. They found that their intervention resulted in a significant increase in ability to discern misinformation with lasting effects if regular misinformation resilience testing occurred over time. Without regular testing they found significant decay over a 2 month period ending in a loss of inoculation effect [37].

## Context and generalisability

There appears to be a significant distinction in how these interventions work between those with preexisting 'healthy' understandings of public health information and those who are the heaviest consumers of misinformation. This was noted in several studies [40–42,46,47,50,54], and in ways that do not initially appear congruent with one another. It is clear this subgroup of heaviest misinformation consumers is impacted differently by many of the interventions included in this review, but that change in impact is not a consistent factor - instead it is an ephemeral variable, difficult to spot and even harder to plan for in study design.

The table below lays out the contexts in which each relevant intervention type was found to be most effective in the groups tested, alongside the groups included within the included studies, relevant findings from the authors regarding context and their intervention, and an overall level of generalisability (Table 3).

**Table 3. Intervention context and generalisability.**

| Inter-vention | Contexts for use | Pilot groups within the studies | Findings on context | Generalis-ability |
|---|---|---|---|---|
| **Accuracy Nudges** | Internet, social media platforms | 1. USA [YouGov] proportional group online (Aslett et al. 2022)<br>2. Students in 'individual chats' on Android devices (Amin et al. 2021)<br>3. USA [MTurk] group online (Dias et al. 2020)<br>4. Kyrgyzstan, India, and USA non-probability samples (Gavin et al. 2022)<br>5. USA [Lucid] proportional group online (Kreps et al. 2022)<br>6. USA [MTurk] group online (Pennycook et al. 2020)<br>7. Kenya & Nigeria facebook users (Offer-Westort et al. 2024)<br>8. Sixteen countries [Qualtrics] (Arechar et al. 2023) | 1. Intervention only works on heaviest misinformation consumers<br>3. Potential for 'Backfire' effect on those most misinformed<br>5. Intervention effectiveness changed with location<br>6. No evidence of 'Backfire' effect | Very high |
| **Pre-bunking** | Universally | 1. USA [YouGov] proportional group online (Amazeen et al. 2022)<br>2. Hong Kong undergraduate students (Jiang et al. 2022)<br>3. USA [Pollfish] proportional group online (Piltch-Loeb et al. 2022) | 1. Intervention only works on those with preexisting healthy attitudes | High |
| **Games** | Youth, digitally literate people, employment mandates | 1. USA [Prolific] proportional group online (Basol et al. 2020)<br>2. China [WeChat] group online (Ma et al. 2023)<br>3. USA [Prolific] proportional group online (Maertens et al. 2021)<br>4. Ghana university students and health workers (Cook et al. 2024) | 1. Intervention works across the political spectrum<br>2. Intervention works well for general public<br>4. intervention works well for the educated and youth | High |
| **Debunking** | Reactively to widely believed misinformation, older people | 1. UK and Brazil Whatsapp users (Vijaykumar et al. 2021)<br>2. Netherlands elderly (Yousuf et al. 2021) | 1. Most effective on older people | Medium |
| **Education** | Difficult to reach communities, communities distrustful of government where peers and individual study might be most effective | 1. USA [Prolific] proportional group online (Agley et al. 2021)<br>2. USA [MTurk] group online (Johnson et al. 2022)<br>3. Older adults in Hong Kong (Fung et al. 2022)<br>4. USA [Prolific] proportional group online (Van Stekelenburg et al. 2021)<br>5. Canada/USA [Prolific] women online (Veletsianos et al. 2022)<br>6. International social media users (Vandormael et al. 2021)<br>7. USA [MTurk] adults online (Abroms et al. 2024)<br>8. Denmark [YouGov] proportional online group (Rasmussen et al. 2022) | 2. Intervention worked best on older and less vaccine-hesitant people<br>4. Intervention caused 'Backfire' effects in conservative Republicans<br>6. Most effective for low baseline knowledge levels | Medium |
| **Message framing** | distrustful communities, those 'bought-in' to conspiracy already, when dealing with political polarisation | 1. Germany non-probability sample (Bender et al. 2023)<br>2. US [MTurk] group online (Iles et al. 2022)<br>3. UK [Lucid] proportional group online (Freeman et al. 2021)<br>4. Australia [Dynata] proportional group online (Li et al. 2024)<br>5. Australia [Qualtrics panels] (Spalvins et al. 2024)<br>6. Mozambique micro-entrepreneurs and household heads (Armand et al. 2024) | 1. Extant framing best for those anxious about vaccination. Intervention framing best for those strongly anti-vaccine<br>2. Emphasising personal benefit more effective on those most vaccine hesitant.<br>3. Emphasising collective benefit creates 'Backfire' effects | Low |
| **Community Engagement** | deprived communities, vulnerable communities, outliers, contexts with significant pre-existing community health networks | 1. American 'Latinx' communities (DeGarmo et al. 2022)<br>2. USA young black adults (Maragh-Bass et al. 2022)<br>3. USA 'essential workers' (Ugarte et al. 2023)<br>4. Tanzania Rukwa region (Msunyaro et al. 2022) | 1. Effective for mitigating health disparities | Very Low |

Eight reviewed studies found insignificant trends in intervention impact between baseline participants and special groups, with several more looking for such trends and finding none. This indicates the specificity of these intervention-types, and that although context and social group could be determinants of intervention effectiveness, such effects are likely to be small. For example, Bender et al. [54] noted that their intervention framing worked best on those already strongly anti-vaccine. Conversely, Johnson et al. [52] found their intervention worked best on those with less vaccine hesitancy, and that those with higher social political conservatism performed worse on knowledge scores. The insignificant trends found in these studies were typically tied to either age, ethnic group, or political ideology as core identities tied to perceptions and experiences of COVID-19 and the public health responses thereto. Political (rightwing/conservative) ideology was noted in many studies as a subgroup of particular importance and was found to coincide with less accurate pre-intervention beliefs [46,50].

Accuracy nudges were tested with participants from the USA, Kyrgyzstan, Kenya, Nigeria, India, and a comprehensive study on 16 countries with findings that suggest that their impact is difficult to predict and changes depending on the context [27]. Dias et al. [41] noted the potential for a 'backfire' effect among those people most bought-in to misinformation, whereas Kreps et al. [42] found no evidence of this effect. Aslett et al. [40] find that their intervention only worked on those who consume the highest levels of misinformation in their participant group and had minimal effect on anyone else. This conflicts with concerns about backfire effects.

Prebunking was tested with participants from online recruiters in the USA and Hong Kong undergraduates. Amazeen et al. [47] found that the intervention only worked on those with preexisting 'healthy' attitudes, meaning those whose beliefs already coincided most closely with legitimate public health messaging. Because this intervention-type is intended to inoculate the 'average' person against misinformation, it only working on those with preexisting 'healthy' attitudes does not reduce the usefulness of prebunking.

Games were tested in the USA, Ghana, and China with proportionally representative online groups. Basol et al. [46] as well as Ma et al. [55] respectively found that the interventions worked across both the political spectrum and the public in general. This indicates high generalisability, particularly with the proportionally representative and relatively large participant cohorts in these studies. However, by the nature of a digital intervention type like games, older people and those with low levels of digital literacy (who are among those most desirable to target for the intervention) may have less desire or ability to play the game.

Debunking was tested in the UK and Brazil among Whatsapp users, and in the Netherlands among the elderly. Interestingly, Vijaykumar et al. [44] found that their intervention was most effective on older people. This indicates that this type of intervention might be most useful among elderly populations and communities. Vijaykumar et al. [44] and Yousuf et al. [45] speculate that perhaps older people have higher baseline trust in governmental messaging and are therefore more open to changing their internalised beliefs based on new information from legitimate sources. By the nature of debunking, it can only be applied reactively to widely believed misinformation, which significantly limits its generalisability.

Education was tested in the USA, Canada, Denmark, Hong Kong, and internationally through social media sharing. Johnson et al. [52] found their intervention worked best on elderly people and those with less hesitancy around COVID-19 vaccination. Similarly, Veletsianos et al. [38] found that their intervention caused a noteworthy 'backfire' effect among conservative US republicans (as the most vaccine-hesitant and 'bought-in' to misinformation already). Vandormael et al. [49] suggested educational interventions might be most effective among populations with a low baseline knowledge level, as their own participant group has relatively high levels of baseline knowledge (although nonetheless the intervention successfully boosted knowledge of COVID-19 prevention). When taken together, these findings indicate that the groups most ideal for this type of intervention are communities with low baseline knowledge of public health information or communities distrustful of government where peer and individual study might be able to penetrate that distrust.

Message framing was tested in Germany [54], the US [36], Mozambique, and the UK [53] all through online interventions testing framed messaging against traditional extant public health informative messaging. Bender et al. [54] found

that extant framing (which typically focuses on collective benefits and informing about vaccination side-effects) worked best for those anxious about vaccination, whereas the intervention framing worked best for those strongly anti-vaccine. Similarly, Freeman et al. [53] found that emphasising personal benefit (the intervention framing) was more effective on the most vaccine-hesitant, whereas emphasising collective benefit (the control/extant framing) was far less effective and even resulted in 'backfire' effects. Together these findings make a strong case for message framing interventions to effectively target those communities most distrustful of government messaging, those most 'bought in' to conspiracy and misinformation already, and the most politically radicalised.

Community engagement was tested in the US among 'Latinx' communities [35], young Black adults [43], and 'essential workers' [34]. By its nature, community engagement is very low generalisability as it is more contextually specific, resource intensive, and time-consuming than any other intervention type. Degarmo et al. [35] found their intervention was successful at mitigating health disparities in the communities they engaged. This suggests community engagement would be most effectively utilised in deprived communities, vulnerable communities, and those areas most difficult to reach for any reason.

## Discussion

The research questions in this study do not have explicit ranked answers, as impact and scalability differ widely across the interventions included in this review. There are tradeoffs in play, between impact and scalability as well as between generalisability and targeted intervention against subgroups of particular importance. Therefore, the key finding from this review is the insufficiency of any one intervention to address the widely varying needs of the many contexts and groups in which misinformation can spread. For further details on contextual fit for different interventions, please refer to the Results sub-section titled: "Context and Generalisability". There is a need for the development of comprehensive packages (each containing multiple interventions) as the core policy recommendation. These packages can pull from the different strengths of each intervention type reviewed to best fit the needs of the relevant communities and contexts within which these packages will be developed. This package approach appears to be gaining momentum, with a higher proportion of newest studies incorporating multiple intervention techniques in their studies. When such a package of multiple interventions is impossible, game-type interventions appear to be an outlier in terms of being highly scalable, impactful, low resource-intensity, and highly generalisable relative to the other intervention-types reviewed. Games are catching and interactive with participants, which could explain the significance found in their effect as an intervention. This interaction element could also lead to durable effects over time, although this is insufficiently studied in the current literature. Furthermore, games have the unique distinction of being fun to play, and to hold the potential to encourage public engagement with the intervention outside of experimental contexts or policies, but simple spread of an enjoyable game to play. For this same reason, games hold the potential to be a very effective subset of education interventions, where, e.g., children in school could play the game as part of their curriculum attempting to build inoculation to misinformation from an early age.

### Politics and partisan bias

Both the theoretical and intervention literatures around COVID-19 misinformation hint at its politically polarising elements yet fail to address this influence head on. It is important to note that this failing is limited to the narrowly confined studies relevant to this study – the public health literature investigating COVID-19 misinformation interventions. The wider literature around misinformation has well-established links to polarisation and politics, particularly in the US. Dispersed throughout the findings and discussions of the included studies are the political elements of COVID-19 misinformation. It is consistently found that political conservatives, particularly in the US, are uniquely vulnerable and bought-in to misinformation and conspiracism [26,57]. This group was found to have its own unique interactions with many of the tested interventions in this review. When this happened, the authors mention this difference and give some speculation as to why that might be the case, but do not investigate this finding further, or seek to use explanations in the wider literature to support

their findings (see [51] for the most comprehensive discussion of this issue in the eligible studies). Additionally, there has been very little work to explicitly begin from this starting point and deep dive into why this might be the case and how interventions might most effectively impact this group. This presents a significant detriment to reaching the stated goal of these interventions - effectively combatting COVID-19 misinformation.

Pennycook et al. [23] is the most influential study included in this review in terms of citation count, references throughout the reviewed studies, and the extent to which their study has been replicated and critiqued within both the studies under review and the wider literature. Within that study they champion the theory that the systemic sharing behaviour of COVID-19 misinformation in our society is "because [people] simply fail to think sufficiently about whether or not the content is accurate when deciding what to share" [23]. Pennycook et al. claim that their findings and this theory indicate that accuracy nudges are not only simple and effective, but the only intervention needed against COVID-19 misinformation. In doing so, they negate the claims of many of the other included studies in this review. This has brought significant criticism against this core idea of what is causing vulnerability to COVID-19 misinformation. If the only issue is a lack of thinking, then accuracy nudges are the obvious intervention. Yet although the findings of Pennycook et al. [23] do suggest the effectiveness of accuracy nudges and the need for interventions that make people think more about their sharing decisions, this 'theory' they promote is insufficiently supported when applied to negating the findings of other studies. Their findings suggest the effectiveness of accuracy nudges, but not the ineffectiveness of other interventions. The alternative proposed answer to what is causing vulnerability to misinformation is partisan bias. This explanation posits that it is not failing to think sufficiently or lower cognitive ability that leads to vulnerability to misinformation, but rather the inherent bias that arises from adherence to political ideology in the context of intense political division and polarisation as is affecting the contemporary United States very deeply but also affects many countries today [58]. This debate on partisan bias vs insufficient thinking punctuates the literature on misinformation, including many of the studies included in this review. This debate continues into the present, with even some of the newest studies promoting accuracy nudges and dismissing other interventions for the same reasons initially proposed by Pennycook et al. [23].

## Limitations

A primary limitation in this review comes from the heterogeneity of the studies and interventions disallowing meta-analysis and other forms of traditional systematic review analysis that rely on similar outcome measures and methodologies within the eligible studies. This limitation is accentuated by the potential for interpretation bias. The interpretation of the data herein is biased by the perspective and worldview of the authors. Additionally, there is limited consistency between realist reviews and limited standards and assessments available to apply to this review. This does not necessarily limit the rigor of the review but makes analysing that rigor and validity more difficult. The development of more and consistent direction and assessments for realist reviews would address this limitation currently present within the method. Lastly, the limited engagement in the intervention literature with theory limits the extent to which theoretical insights can be drawn from this study.

## Future research directions

Although a variety of interventions tested in the studies herein found success in the short term, in the long-term it is impossible to avoid the urgent need for mass-scale education on digital literacy if the goal is to make a population as resilient as possible against misinformation. Given the variability in reporting backfire effects and subgroup-specific outcomes, future interventions should prioritize the development of tailored packages that account for these factors. This should be informed by emerging literature, including ongoing studies (such as the author's upcoming work), which will further investigate the role of backfire effects and subgroup-specific outcomes. Future research in this direction is pivotal, with experiment-groups in classrooms a clear next step. Additionally, future research on how to address the political difficulties in implementing such a wide-scale intervention is required.

Out of all intervention-types reviewed, games appear to create the highest impact while still being highly scalable and resource-inexpensive, with the potential for longevity in the right conditions [37]. Relative to the other intervention-types, games scores maximally in terms of impact on participants, while still being relatively high on scalability. Future research in this direction is needed to refine and test these results. Longitudinal testing is an obvious follow-up to gain insight into durability of inoculation effect.

Additional areas for future research include: 1) theoretical research into how to build a resilient population and how to address vulnerability to misinformation systemically versus individually; 2) the role of politics and partisan bias in the functioning of these interventions; 3) where misinformation comes from and who gains from it; 4) the role of political polarisation and radicalisation in vulnerability to and the spread of misinformation, both in the United States and globally; 5) standardisation measures or frameworks for consistent evaluation of interventions to allow for meta-analysis and quantitative comparison; and 6) policy-focused principles for the development of intervention packages to guide policymakers.

## Conclusions

This review included 35 studies of interventions combatting COVID-19 misinformation. The interventions reviewed varied widely in terms of scalability, resource intensity, impact on participants, the contexts within which each best works, the people onto whom the interventions will have greatest effect, and research quality. The tests performed in the included studies hold rich contributions toward better understanding how misinformation functions, how veracity judgement occurs in individuals and communities, and which interventions work best in which contexts and for whom. COVID-19 showed precisely how harmful and deadly misinformation can be, and what a public health threat it can represent. In this fight against systemic misinformation in our society, a final takeaway from this review is the need for acknowledgement of misinformation as a societal and systemic issue that requires significant investment and time to resolve, if resolution is possible.

## Supporting information

**S1 File. PRISMA checklist.**
(DOCX)

**S2 File. Pre-registered protocol.**
(PDF)

**S1 Table. Quality assessment table.**
(DOCX)

**S2 Table. Article eligibility tracking table.**
(XLSX)

## Acknowledgments

We want to thank Katie Goddard from the Primary Care and Public Health department of the Brighton and Sussex Medical School for her help in duplicating and affirming the deselection and qualitative coding in this study.

This study took place as part of the PhD candidacy of Robert Dickinson at the University of Sussex. Non-financial support came from project supervisors Dominique Mackowski, Harm Van Marwijk, and Elizabeth Ford. Additionally, Katie Goddard performed the role of deselection replication as laid out in the methodology.

## Author contributions

**Conceptualization:** Robert Dickinson.

**Data curation:** Robert Dickinson.

**Formal analysis:** Robert Dickinson.

**Investigation:** Robert Dickinson.

**Methodology:** Robert Dickinson, Elizabeth Ford, Harm van Marwijk.

**Project administration:** Robert Dickinson.

**Software:** Robert Dickinson.

**Supervision:** Dominique Makowski, Elizabeth Ford, Harm van Marwijk.

**Validation:** Dominique Makowski.

**Visualization:** Robert Dickinson.

**Writing – original draft:** Robert Dickinson.

**Writing – review & editing:** Robert Dickinson, Dominique Makowski, Elizabeth Ford, Harm van Marwijk.

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
