## [Decision Letter · Decision Letter 0]

12 Feb 2025

PONE-D-24-46282Effectiveness of digital health interventions against COVID-19 misinformation: a systematic realist review of intervention trialsPLOS ONE

Dear Dr. Dickinson,

Thank you for submitting your manuscript to PLOS ONE. After careful consideration, we feel that it has merit but does not fully meet PLOS ONE’s publication criteria as it currently stands. Therefore, we invite you to submit a revised version of the manuscript that addresses the points raised during the review process. Please find the reviewers comments below this email.

We look forward to receiving your revised manuscript.

Kind regards,

Osmond Ekwebelem

Academic Editor

PLOS ONE

2. As required by our policy on Data Availability, please ensure your manuscript or supplementary information includes the following: 

Reviewers' comments:

Reviewer's Responses to Questions

**Comments to the Author**

1. Is the manuscript technically sound, and do the data support the conclusions?

Reviewer #1: No

Reviewer #2: Yes

Reviewer #3: Yes

2. Has the statistical analysis been performed appropriately and rigorously? 

Reviewer #1: N/A

Reviewer #2: N/A

Reviewer #3: N/A

3. Have the authors made all data underlying the findings in their manuscript fully available?

Reviewer #1: No

Reviewer #2: Yes

Reviewer #3: Yes

4. Is the manuscript presented in an intelligible fashion and written in standard English?

Reviewer #1: Yes

Reviewer #2: Yes

Reviewer #3: Yes

5. Review Comments to the Author

Reviewer #1: Thank you for the opportunity to review this manuscript. While a realist review of interventions against COVID-19 misinformation is timely, I have significant concerns about the theoretical framework employed, which ultimately impacts the validity of the conclusions. Therefore, I recommend rejecting the paper in its current form.

Regarding the theoretical concerns, the introduction has many affirmations without proper citations. I won’t name each of them but I think it is remarkable that the first page includes just one citation. This of course has ramifications, and the idea that Precovid interventions against misinformation were based only in fact-checking and nudges show to some extent unfamiliarity with a literature that is broad in topics like vaccine hesitancy and HIV misinformation. Also related to this point, a realist review should clarify the underlying mechanisms that explain how an intervention produces its effects and in which contexts these effects take place. My impression is that although the paper explains the different kinds of interventions evaluated, it do not provide a theoretical framework to understand the different mechanisms and contexts that produce different outcomes.

The lack of a robust theoretical framework hinders the interpretation of findings. One example of this is the conclusion about addressing polarization and politics bias to understand vulnerability towards misinformation. The relationship of polarization and misinformation is now commonplace i the US literature, but it is not clear how this review raised flags about it. Moreover, it is not clear if in the literature this was a global finding or it is just applicable to the US case. Finally, the conclusions that no intervention is sufficient to counter misinformation and that games interventions are more effective should be further elaborate, to unravel which interventions are proper for which contexts and populations, as that is the ultimate goal of a realist review.

Finally, author should take into account that PlosONE have specific criteria for publication of systematic reviews (https://journals.plos.org/plosone/s/criteria-for-publication and https://journals.plos.org/plosone/s/submission-guidelines#loc-systematic-reviews-and-meta-analyses). The first problem I find with this submission is that while PRISMA flow diagram and checklist are included in the submission, the checklist items were not developed as a standalone document, but have a “location where item is reported” that do not seem to correspond with a page number of the manuscript. Also, the protocol submitted with PROSPERO should be included as supporting information, and it would be valuable to address RAMSES II standards for realist reviews. Finally, related to PlosONE criteria for publication, the manuscript does not follow the standard structure of the journal publications. For example, the methods section is presented as the last section, while in PlosONE it usually is after the introduction. This hinders comprehension, and seems like an editing error for the Study characteristic section mentions information given “before” that is actually presented at the end of the document.

Despite these concerns, I believe this review has the potential to make a valuable contribution to the understanding of misinformation interventions. I encourage the authors to significantly revise their manuscript before resubmitting for publication. I hope this feedback proves helpful in your research process.

Reviewer #2: Dear Robert,

Thank you for submitting this systematic review on interventions against COVID-19 misinformation. This is an important contribution to public health research, addressing a critical challenge that has had far-reaching consequences. While the review is commendable in its scope and aims, I have identified some areas that could benefit from further attention and refinement.

1.Scope of the Review:

Is the review exclusively focused on digital health interventions (i.e., internet/electronic device-based approaches)? While many of the interventions discussed in the paper include digital components, others—such as community engagement initiatives involving health promoters and outreach activities—do not appear to have a clear digital aspect. Similarly, some interventions categorized as “educational” or “message framing” may not inherently fall under the umbrella of digital health. As such, the current title of the article may inadvertently lead readers to believe that the review is limited to digital interventions. I recommend considering a title that better reflects the broader scope of the interventions analyzed.

2. Flow Diagram Discrepancy:

The flow diagram indicates that 65 studies were assessed for eligibility, with 45 excluded for reasons such as lack of intervention or relevance to COVID-19. However, the original review ultimately included 26 studies, which does not align with the arithmetic presented (65 - 45 = 20). Could you clarify or revise the flow diagram to ensure consistency and transparency in the study selection process?

3.Clarity and Focus on Intervention Types:

• Despite the review highlighting the eight types of interventions, the categorization and definitions of these types could be clearer. It is not always evident what distinguishes certain categories, and some appear to overlap conceptually. A clearer framework or typology of interventions would enhance the reader's understanding of the findings.

• The emphasis on game-type interventions is interesting, but it would be helpful to discuss why these interventions performed better in terms of impact and scalability. Are there specific features or mechanisms that make them more effective?

4. Quality Assessment:

While the review mentions quality assessment, it is not clear how the quality of the included studies influenced the synthesis of findings. The review indicates that no study was excluded based on poor quality. Were lower-quality studies given less weight, or were their limitations explicitly considered in the analysis? Clarifying this would enhance the rigor of the review.

5. Heterogeneity of Outcome Measures:

The review notes high heterogeneity in outcome measures and methodologies across the included studies, which complicates comparisons. While this is an important observation, the review could delve deeper into how future studies might standardize these measures or propose a framework for consistent evaluation of interventions.

6. Backfire Effects and Subgroup Analysis:

The intermittent reporting of backfire effects and subgroup-specific outcomes is a critical limitation. It would strengthen the review to provide more detail on the implications of these gaps. For instance, how might the variability in reporting backfire effects impact the development of tailored intervention packages?

7. Expansion of Realist Review Approaches:

The review mentions contributing to the expansion of realist review methodologies but does not fully explain how this contribution is achieved. Expanding on this would strengthen the review’s methodological impact.

8. Contextual Considerations and Policymaker Recommendations:

The recommendation for policymakers to create tailored intervention packages is valuable. However, the review could benefit from more specific guidance on how to operationalize these packages. Are there concrete examples or evidence-based principles that could guide the tailoring process for different contexts and subgroups?

In summary, this is a highly relevant and well-intentioned review that provides valuable insights into a pressing public health issue. Addressing the above points would further enhance the clarity, rigor, and applicability of the findings, making this work even more impactful.

Best regards,

Reviewer #3: I have with interest read the manuscript which aim to summarize research on covid-19 misinformation information. I am not a social or communication scientist, but working with covid-19, also in a public health perspective. From what I understand the research field of misinformation is new but rapidly expanding. I find the subject of this review of high importance. The manuscript is written as a realist review, trying to explore what works and for whom. The approach gives the authors a large amount of freedom how to analyze and present the findings.

A lot of words and phrases are a new to me. As a reader I would be help with a figure or “fact box” explaining all interventions available, and which are included in the analysis and together with a short example – to but everything in a context. As of now I have difficulty finding out what for example debunking really means, as some interventions are mentioned in the introduction and some are not.

From what I understand, even the definition of the word misinformation is controversial, and perhaps the author’s definition should be stated in the manuscript.

I would also appreciate a discussion of how the research concerning covid-19 misinformation differ from other misinformation research.

The references are used in an order I don’t often se, with references in the middle of a sentence and in some places an appropriate reference is missing (line 373 for example). I recommend looking over, and perhaps expanding the references.

6. PLOS authors have the option to publish the peer review history of their article (what does this mean? ). If published, this will include your full peer review and any attached files.

**Do you want your identity to be public for this peer review?** For information about this choice, including consent withdrawal, please see our Privacy Policy .

Reviewer #1: No

Reviewer #2: **Yes: ** Tinkhani Mbichila

Reviewer #3: No

---

## [Author Response · Author response to Decision Letter 1]

11 Mar 2025

Response to Reviewers: Interventions for Combating COVID-19 Misinformation: A Systematic Realist Review

Dear Reviewers,

We sincerely appreciate your time and detailed feedback on our manuscript. We have carefully addressed all concerns raised and made substantial revisions to enhance the clarity, rigor, and methodological transparency of our review. Below, we outline our responses to each point and the corresponding changes made to the manuscript.

***please note: line numbers given here refer to the manuscript with tracked changes, and will not correspond to the clean copy***

Journal Requirements

1. Style Requirements

Comment: “Please ensure that your manuscript meets PLOS ONE's style requirements, including those for file naming. The PLOS ONE style templates can be found at https://journals.plos.org/plosone/s/file?id=wjVg/PLOSOne_formatting_sample_main_body.pdf and https://journals.plos.org/plosone/s/file?id=ba62/PLOSOne_formatting_sample_title_authors_affiliations.pdf”

Response: We have ensured that the manuscript meets PLOS ONE's style requirements, including those for file naming. Thank you for the helpful links!

2. Table of all studies

Comment: “As required by our policy on Data Availability, please ensure your manuscript or supplementary information includes the following:

A numbered table of all studies identified in the literature search, including those that were excluded from the analyses. For every excluded study, the table should list the reason(s) for exclusion. If any of the included studies are unpublished, include a link (URL) to the primary source or detailed information about how the content can be accessed.”

Response: A numbered table of all studies identified in the literature search has been constructed. Please note: duplicated studies are shown only once in the table for clarity purposes. Reasons for exclusion are given. All included studies were published.

3. Data extraction table

Comment: “As required by our policy on Data Availability, please ensure your manuscript or supplementary information includes the following:

A table of all data extracted from the primary research sources for the systematic review and/or meta-analysis. The table must include the following information for each study: Name of data extractors and date of data extraction; Confirmation that the study was eligible to be included in the review. All data extracted from each study for the reported systematic review and/or meta-analysis that would be needed to replicate your analyses.”

Response: We acknowledge the importance of including a table with extracted data from the primary research sources. However, due to the nature of this review, where all text from each included study was thoroughly iterated through and qualitatively coded, compiling a full table of all extracted data would be overly extensive and not representative of the iterative process used. Instead, we refer reviewers to the Methods section, where the data extraction and coding process is described in detail. This ensures clarity and transparency while maintaining the practicality of the submission. In this instance, a full data extraction table would include all prose from all included studies.

4. Quality assessment table

Comment: “As required by our policy on Data Availability, please ensure your manuscript or supplementary information includes the following:

If applicable for your analysis, a table showing the completed risk of bias and quality/certainty assessments for each study or outcome. Please ensure this is provided for each domain or parameter assessed. For example, if you used the Cochrane risk-of-bias tool for randomized trials, provide answers to each of the signalling questions for each study. If you used GRADE to assess certainty of evidence, provide judgements about each of the quality of evidence factor. This should be provided for each outcome.”

Response: Table showing completed risk of bias assessments for each study has been shifted from Appendices to a supplementary information file per PLOS ONE guidelines.

5. Missing data

Comment: “An explanation of how missing data were handled.”

Response: Although this review does not claim perfect capture of all data relevant to this area, within the confines of the study as laid out in the methodology no missing data was encountered.

Reviewer 1

1. Citations

Comment: “Regarding the theoretical concerns, the introduction has many affirmations without proper citations. I won’t name each of them but I think it is remarkable that the first page includes just one citation. This of course has ramifications, and the idea that Precovid interventions against misinformation were based only in fact-checking and nudges show to some extent unfamiliarity with a literature that is broad in topics like vaccine hesitancy and HIV misinformation.”

Response: We have added ~20 citations to support key claims not just in the introduction but throughout the paper. Furthermore, we have toned down the referenced statement on precovid interventions and added in additional prose expanding upon the pre-covid misinformation intervention literature, beginning line 67 now reading: “The pre-COVID misinformation intervention landscape appears to focus on fact-checking [10–14]. This is not to suggest that this literature exclusively focused on fact-checking, as a rich literary body nonetheless existed examining with depth and alacrity understandings of the effectiveness and appeal of misinformation based in cognitive ability, emotional appeal, partisanship, sensationalism, and fear-mongering[15–17].”

2. Theoretical Framework

Comment: “a realist review should clarify the underlying mechanisms that explain how an intervention produces its effects and in which contexts these effects take place. My impression is that although the paper explains the different kinds of interventions evaluated, it do not provide a theoretical framework to understand the different mechanisms and contexts that produce different outcomes. The lack of a robust theoretical framework hinders the interpretation of findings.”

Response: We agree with this insightful suggestion, and therefore have added a theoretical framework as a new section to the manuscript. This section beginning line 115 reads as follows:

“Theoretical Framework

This study takes the theoretical framing of a realist review, focusing on uncovering the mechanisms that explain how interventions produce effects in specific contexts, working to ensure that findings can be generalized or applied to similar circumstances. These mechanisms function as the underlying processes that drive change in an intervention. They could be cognitive, as is the case for accuracy nudge interventions, where a brief reminder provides a cognitive ‘nudge’ to temporarily boost cognition when clicking on a link or news article. They could be social, as is the case for community engagement interventions, where extended and deep interaction with a community works to identify and slowly change misinformed group narratives. They could also be behavioural, as is the case for game interventions, where the act of playing a game itself is the intervention, and the interaction between player and game functions to educate and inoculate against misinformation. Table 1a in the results section provides a description of the mechanism at work within each included study for this review.

The other key piece of a realist review puzzle, contexts, are the environmental, cultural, economic, or other situational factors that influence how, why, where, and when interventions are effective. For some interventions like accuracy nudges, their use is limited to digital spaces, fitting contexts like social media platforms. For others like educational or message framing interventions, fitting contexts for use might include difficult to reach or distrustful communities. For debunking interventions, context becomes a question of time and popularity, best fitting situations where misinformation has already emerged and spread. Table 2 in the results section, alongside the wider ‘Context and Generalisability’ subsection of the results explains in greater detail the contextual nature of the interventions included in this review, as well as ranking them by generalisability.

By together clarifying the mechanisms and contexts for the interventions included, this review provides a deeper understanding of how the interventions operate and offers a theoretical contribution to realist review methods in public health and psychology. By utilising a realist review approach in the study of misinformation, where little such examination currently exists, this study contributes to the expansion of realist review approaches. Similarly, as both realist reviews and misinformation are quickly growing and relatively newly expanding areas of literature, this approach offers new insights into the study of misinformation interventions, particularly in the case of COVID-19. This review also highlights the importance of context in the countering of misinformation. By demonstrating the limited generalisability of many interventions, it contributes to the ongoing refinement of misinformation interventions to best allow bespoke policy packages of interventions to be developed best fitting the needs of the communities involved.”

3. Polarization & Global Context

Comment: “One example of this is the conclusion about addressing polarization and politics bias to understand vulnerability towards misinformation. The relationship of polarization and misinformation is now commonplace i the US literature, but it is not clear how this review raised flags about it. Moreover, it is not clear if in the literature this was a global finding or it is just applicable to the US case.”

Response: We found these comments helpful when considering how to justify and expand the discussion section. Firstly, we added a more explicit statement considering the wider misinformation literature compared to the narrower COVID-19 misinformation intervention literature (the focus of this study). New prose beginning line 505 reads as follows: “It is important to note that this failing is limited to the narrowly confined studies relevant to this study – the public health literature investigating COVID-19 misinformation interventions. The wider literature around misinformation has well-established links to polarisation and politics, particularly in the US.” We believe this addresses many of the concerns of the reviewer, but further agree that there is a continued question as to the global applicability of political polarization outside of the US, and therefore have amended the final direction for future research in the discussion section, beginning line 569 now reading: “4) the role of political polarisation and radicalisation in vulnerability to and the spread of misinformation, both in the United States and globally”

4. Justification for Conclusions

Comment: “Finally, the conclusions that no intervention is sufficient to counter misinformation and that games interventions are more effective should be further elaborate, to unravel which interventions are proper for which contexts and populations, as that is the ultimate goal of a realist review.”

Response: We have expanded our discussion to highlight why game-based interventions perform well in terms of both impact and scalability. New prose beginning line 495 reads: “Games are catching and interactive with participants, which could explain the significance found in their effect as an intervention. This interaction element could also lead to durable effects over time, although this is insufficiently studied in the current literature. Furthermore, games have the unique distinction of being fun to play, and to hold the potential to encourage public engagement with the intervention outside of experimental contexts or policies, but simple spread of an enjoyable game to play. For this same reason, games hold the potential to be a very effective subset of education interventions, where e.g. children in school could play the game as part of their curriculum attempting to build inoculation to misinformation from an early age.”

Additionally, we provide clearer guidance on which interventions may be best suited for different target populations by directing readers to the Results section devoted thereto. This prose beginning line 486 reads: “For further details on contextual fit for different interventions, please refer to the Results sub-section titled: “Context and Generalisability”.”

5. Formatting & PRISMA Checklist

Comment: “Finally, author should take into account that PlosONE have specific criteria for publication of systematic reviews (https://journals.plos.org/plosone/s/criteria-for-publication and https://journals.plos.org/plosone/s/submission-guidelines#loc-systematic-reviews-and-meta-analyses). The first problem I find with this submission is that while PRISMA flow diagram and checklist are included in the submission, the checklist items were not developed as a standalone document, but have a “location where item is reported” that do not seem to correspond with a page number of the manuscript. Also, the protocol submitted with PROSPERO should be included as supporting information, and it would be valuable to address RAMSES II standards for realist reviews. Finally, related to PlosONE criteria for publication, the manuscript does not follow the standard structure of the journal publications. For example, the methods section is presented as the last section, while in PlosONE it usually is after the introduction. This hinders comprehension, and seems like an editing error for the Study characteristic section mentions information given “before” that is actually presented at the end of the document.”

Response: We have restructured the manuscript to align with PLOS ONE’s formatting guidelines, including moving the Methods section earlier in the document. These adjustments were not made initially, in accordance with PLOS ONE’s guidance that formatting requirements could be waived until after the revisions stage. The PRISMA checklist has been revised for accuracy and is now provided as a standalone document with page numbers clearly indicated. As PLOS ONE requires adherence to the PRISMA standards rather than the RAMSES II standards, this study has been focused on meeting the PRISMA requirements for systematic review publication. However, we would like to note that the study follows best practices for realist reviews and has critically engaged with the RAMSES II guidelines for authors.

Reviewer 2

We would like to begin by thanking Reviewer 2 for their helpful comments, and particularly the comprehensive and ordered way they laid out their comments. This made for an exceptionally straightforward response and was greatly appreciated.

1. Scope of the Review

Comment: “Is the review exclusively focused on digital health interventions (i.e., internet/electronic device-based approaches)? While many of the interventions discussed in the paper include digital components, others—such as community engagement initiatives involving health promoters and outreach activities—do not appear to have a clear digital aspect. Similarly, some interventions categorized as “educational” or “message framing” may not inherently fall under the umbrella of digital health. As such, the current title of the article may inadvertently lead readers to believe that the review is limited to digital interventions. I recommend considering a title that better reflects the broader scope of the interventions analyzed.”

Response: We have revised the title to "Interventions for Combating COVID-19 Misinformation: A Systematic Realist Review" to better reflect the broader scope of interventions included.

2. Flow Diagram Discrepancy

Comment: The flow diagram indicates that 65 studies were assessed for eligibility, with 45 excluded for reasons such as lack of intervention or relevance to COVID-19. However, the original review ultimately included 26 studies, which does not align with the arithmetic presented (65 - 45 = 20). Could you clarify or revise the flow diagram to ensure consi

---

## [Editor Report · Decision Letter 1]

12 Mar 2025

Interventions for combating COVID-19 misinformation: a systematic realist review

PONE-D-24-46282R1

Dear Mr. Dickinson,

We’re pleased to inform you that your manuscript has been judged scientifically suitable for publication and will be formally accepted for publication once it meets all outstanding technical requirements.

Kind regards,

Osmond Ekwebelem

Academic Editor

PLOS ONE
---

## [Editor Report · Acceptance letter]

PONE-D-24-46282R1

PLOS ONE

Dear Dr. Dickinson,

I'm pleased to inform you that your manuscript has been deemed suitable for publication in PLOS ONE. Congratulations! Your manuscript is now being handed over to our production team.

Kind regards,

on behalf of

Dr. Osmond Ekwebelem

Academic Editor

PLOS ONE